# Integrated Microbiome and Metabolomics Analysis of the Effects of Dietary Supplementation with Corn-Steep-Liquor-Derived *Candida utilis* Feed on Black Pigs

**DOI:** 10.3390/ani14020306

**Published:** 2024-01-18

**Authors:** Huiyu Qi, Ruqi Wang, Chuanqi Wang, Rui Wang, Jinglin Shen, Hengtong Fang, Jing Zhang

**Affiliations:** College of Animal Sciences, Jilin University, Changchun 130062, China

**Keywords:** *Candida utilis* feed, growth performance, fecal microbiota, plasma metabolism, piglet

## Abstract

**Simple Summary:**

Yeast protein feeds have the potential to replace other protein source feeds. Accordingly, in this paper, we used the industrial and agricultural byproducts of glucose master liquor and corn steep liquor as substrates, and utilized *Candida utilis* for fermentation. Firstly, the medium components of yeast protein feed were optimized, and then the optimized yeast protein feed was fed to weaned Dongliao black piglets; the feeding effect was analyzed using 16S rDNA and metabolomics. We found that *Candida utilis* feed has no adverse effects on the growth performance of the pigs and that it improves blood lipids and other indicators. A total of 10 plasma metabolites were correlated with 27 genera of bacteria, as revealed by the results of the combined multi-omics analysis, which were mainly enriched in the pathways of primary bile acid metabolism, histidine metabolism, and tryptophan metabolism. The results show that *Candida utilis* feed can maintain the growth performance of the pigs and enhance their immunity and maintenance of gut flora homeostasis; this likely occurs through amino acid metabolism.

**Abstract:**

In this experiment, glucose master liquor and corn steep liquor were used as carbon and nitrogen sources, and *Candida utilis* was used as a strain to ferment yeast feed. The OD value and number of yeast cells were used as response values to optimize the medium components of the yeast feed through a response surface methodology. The optimal medium components were a glucose master liquor concentration of 8.3%, a corn steep liquor concentration of 1.2%, and a KH_2_PO_4_ concentration of 0.14%. Under this condition of fermentation, the OD value was 0.670 and the number of yeast cells was 2.72 × 10^8^/mL. Then, we fed *Candida utilis* feed to Dongliao black piglets, and the effects of the yeast feed on the piglets’ growth performance, fecal microbiota, and plasma metabolic levels were investigated through 16S rDNA sequencing and metabolomics. In total, 120 black piglets with an average initial weight of 6.90 ± 1.28 kg were randomly divided into two groups. One group was fed the basic diet (the CON group), and the other was supplemented with 2.5% *Candida utilis* add to the basic diet (the 2.5% CU group). After a pre-feeding period, the formal experiments were performed for 21 days. The results showed that the addition of *Candida utilis* to the diet did not affect growth performance compared with the control group. Meanwhile, no significant differences were observed in the serum biochemical indices. However, piglets in the 2.5% CU group had a significantly altered fecal microbiota, with an increased abundance of *Clostridium_sensu_stricto_1*, *Lactobacillus*, and *Muribaculaceae_unclassified*. Regarding the plasma metabolome, the 12 differential metabolites detected were mainly enriched in the histidine, tryptophan, primary bile acid, and caffeine metabolic pathways. Regarding the integrated microbiome–metabolome analysis, differential metabolites correlated with fecal flora to variable degrees, but most of them were beneficial bacteria of *Firmicutes*. Collectively, dietary *Candida utilis* feed had no adverse effect on growth performance; however, it played an important role in regulating fecal flora and maintaining metabolic levels.

## 1. Introduction

Protein in the diet is an essential nutrient for animal growth and plays a vital role in gut health. At present, the structure of feed formulation in China is dominated by corn–soybean meal; however, the excessive dependence on imported soybean meal is detrimental to the economic efficiency of farming. In this context, unconventional protein feed is not yet fully utilized, and there seems to be great potential in developing a new type of protein source as soon as possible to alleviate the shortage of protein sources. Scientists have found that yeast feed offers an attractive alternative source to replace other protein sources; yeast feed is not only easily digested and absorbed but also has a balanced amino acid profile that provides animals with a rich source of nutrients [1].

Some studies have found that, when yeast feed replaces a portion of fishmeal or soybean meal, it can help in maintaining animal growth performance [2,3]. Piglets often face challenges after weaning including diarrhea, low feed intake, and weight loss, which can cause serious damage to gut health and function. Fortunately, the addition of yeast feed can alleviate piglets’ post-weaning stress [4,5]. The potential for the use of yeast feed to supplement deficient nutrients has become increasingly recognized, since yeast has a positive impact on animal health and nutrition, regulating the intestinal flora and stimulating the immune system to prevent the growth of pathogenic bacteria [6]. Currently, the strains used to produce yeast feed are *Saccharomyces cerevisiae*, *Candida utilis*, *Pichia*, and so on, which have functions similar to probiotics and are not only beneficial to human beings but may also play an important role in the nutrition and health of animals. *Candida utilis* is rich in protein and amino acids and its cell-wall-derived mannan-oligosaccharides and glucans are beneficial to piglet growth and intestinal health as feed additives [7]. Furthermore, another advantage of adopting *Candida utilis* is that industrial and agricultural wastes may be utilized as substrates to ferment edible proteins for application in livestock farming [8,9,10,11]. In this way, different feeds fermented with *Candida utilis* and yeast-containing products may provide different nutritional functions to animals as nutritional sources or probiotics [12,13]. However, in order to produce feeds with high protein contents, fermentation parameters, such as optimization of medium composition, culture conditions, need to be continuously optimized to achieve the desired purpose. The response surface methodology is widely used in process optimization studies since it can allow researchers to evaluate the interactions between factors, quickly identify the relationships between factors, and response values and conclude the best optimization parameters [14,15]. Based on the properties of *Candida utilis*, we selected industrial waste, glucose master liquor, and a corn byproduct, corn steep liquor, as fermentation substrates to produce forage protein feed; this approach not only turns waste into a valuable product but is also a step in the right direction in tackling the problem of environmental pollution.

Overall, this study aimed to analyze the effect of *Candida utilis* feed on Dongliao black piglets. We first optimized the medium components of *Candida utilis* feed to achieve the maximum content of yeast cells in the feed, as achieved through a response surface methodology. Then, we added the optimized *Candida utilis* feed to the diets to observe its effects on the growth performance, intestinal flora, and metabolic level of weaned piglets, as determined using 16S rDNA combined with non-targeted metabolomics. In this way, we sought to determine whether it can be used as a novel protein source, thereby providing a theoretical basis for its future application in the aquaculture industry.

## 2. Materials and Methods

Glucose master liquor and corn steep liquor were obtained from Jilin Ruisheng Technology Co. (Songyuan, China). The yeast strain *Candida utilis* CICC1801 was purchased from China Center of Industrial Culture Collection (Beijing, China). Glucose, agar, yeast paste, peptone, yeast extract, ammonium sulfate (NH4)_2_SO_4_, magnesium sulfate (MgSO_4_), and potassium dihydrogen phosphate (KH_2_PO_4_) were purchased from Sinopharm Chemical Reagent Co., Ltd. (Shanghai, China). The entire feeding experiment was conducted at the Agricultural Experimental Base of Jilin University (Changchun, China), and all animals were treated in accordance with the Ethics Committee of Jilin University.

### 2.1. Culture Medium

The activated culture medium used was a Yeast Extract Peptone Dextrose Medium (YPD); it contained 20 g of glucose, 20 g of peptone, 10 g of yeast paste, and 20 g of agar in 1 L of deionized water. The initial seed medium contained 50 g of glucose, 20 g of yeast extract, 2 g of potassium dihydrogen phosphate, 1 g of magnesium sulfate, and 1 g of ammonium sulfate in 1 L of deionized water, pH 5.5. In addition to these, the slant medium contained 15 g of agar. All culture media were sterilized at 121 °C for 21 min.

### 2.2. Methods for the Determination of Strain Indicators

Strain activation: The Candida utilis strain preserved on the slant was taken out and inoculated with the YPD medium using the dilution coating method and incubated in an incubator at 30 °C for 48 h for activation. Then, to observe the colony growth status, a single colony with good growth was taken, inoculated in the slant medium, incubated in the incubator at 30 °C for 48 h and shaken in a bottle to make the seed culture after it grew into moss.

Preparation of seed liquids: A ring of Candida utilis was selected from the slant moss and inoculated in a 500 mL triangular flask containing 100 mL of seed medium, incubated for 20 h at 30 °C and shaken at 160 rpm/min, and stored at 4 °C for the determination of each index in the subsequent test.

Determination of the growth curve: Samples were taken every 2 h during seed cultivation, and the absorbance value was measured at 600 nm using a spectrophotometer after diluting 20 times with distilled water. A growth curve of Candida utilis was plotted with time as the horizontal coordinate and the absorbance value as the vertical coordinate.

The optical density (OD) value was determined using a spectroscopic method (λ = 600 nm) [16], and the number of yeast cells was determined using a counting chamber [17].

### 2.3. Single-Factor Experiment Design of Culture Medium Components

Carbon source concentration: Using the seed medium as a reference, five groups of glucose master liquor concentrations of 2%, 4%, 6%, 8%, and 10% were set as the carbon source; other components were unchanged; the pH was adjusted to 5.5; the strain inoculum was 5%; and fermentation was carried out in a shaking bed at 160 rpm for 20 h at 30 °C. The OD value and number of yeast cells of the fermentation liquid were used as evaluation indexes to initially select the appropriate amount of glucose master liquor to be added.

Nitrogen source concentration: The seed medium was used as a reference; the glucose master liquor concentration was 8%; five groups of corn steep liquor concentrations of 1%, 2%, 3%, 4%, and 5% were set as the nitrogen source; other components were unchanged; the pH was adjusted to 5.5, and the culture conditions remained unchanged. The OD value and number of yeast cells of fermentation liquid were used as evaluation indexes to initially select the appropriate amount of corn steep liquor to be added.

Inorganic salt concentration: Using the seed medium as a reference, the glucose master liquor concentration was 8%; the corn steep liquor concentration was 1%; five groups of KH_2_PO_4_ concentrations of 0.05%, 0.1%, 0.15%, 0.2%, and 0.25% were set; other components were unchanged; the pH was adjusted to 5.5, and the culture conditions remained unchanged. The OD value and number of yeast cells of fermentation liquid were used as evaluation indexes to initially select the appropriate amount of KH_2_PO_4_ to be added.

### 2.4. Response Surface Methodology

The Box–Behnken design and the response surface methodology (RSM) were used to further optimize the medium components. The variables selected using the single-factor experiment were designed as three levels (low, middle, and high) and coded as −1, 0, and 1, respectively. According to the coding design, Design-Expert software generated a 3-factor, 3-level, 17-run experimental design scheme, using the OD value and number of yeast cells as response values to optimize the medium components.

### 2.5. Animals, Diets, and Experimental Design

A total of 120 Dongliao black pigs, acquired from the Agricultural Experimental Base of Jilin University, were weaned at 28 days. Using the optimized medium components for actual fermentation production, the fermentation liquid was enzymatically digested by adding alkaline protease. The enzymatic liquid was filtered to obtain the concentrated broth, and then spray-dried, resulting in the finished product of the *Candida utilis* feed. One group was fed the basic diet (the CON group) supplemented with *Saccharomyces cerevisiae*, and the experimental group was supplemented with 2.5% *Candida utilis* feed. The piglets had free access to feed and drinking water at all times. On the last day of the experiment, each pig’s body weight (BW) and feed consumption were recorded, and these data were used to calculate the growth performance indexes ADG, ADFI, and FCR. All diets were fortified to meet or exceed the NRC (2012) nutritional requirements for piglets (Table 1).

### 2.6. Sample Collection

On days 19–21 of the trial, collected feces were immediately stored in liquid nitrogen or at −80 °C for subsequent analysis. Blood samples were collected via a 10 mL jugular vein puncture and promptly centrifuged at 3500× *g* for 10 min to collect serum. These samples were used for serum biochemical assays and a subsequent histologic analysis.

### 2.7. Serum Biochemical Indicators Analysis

The blood samples were incubated in a 37 °C water bath for 10 min, centrifuged at 3000 r/min for 15 min to collect serum, and stored at −80 °C. Serum biochemical indicators included triglycerides (TGs), total cholesterol (TC), total bilirubin (T-BIL), direct bilirubin (D-BIL), γ-glutamyltransferase (GGT), high-density lipoprotein (HDL), low-density lipoprotein (LDL), aspartate aminotransferase (AST), alanine aminotransferase (ALT), and alkaline phosphatase (ALP), which were measured using corresponding commercial kits (Medicalsystem Biotechnology Co., Ltd., Ningbo, China) and an automatic biochemical analyzer (MS-880B, Medicalsystem Biotechnology Co., Ltd., Ningbo, China).

### 2.8. DNA Extraction and Purification and 16S rDNA Amplification Data Analysis

After the fecal samples collected from each pig were completely dissolved, according to the manufacturer’s instructions, DNA from the different samples was extracted using an E.Z.N.A. ^®^Stool DNA Kit (D4015, Omega, Inc., Norwalk, CT, USA). Nuclease-free water was used as a blank control. The DNA quality was tested on the machine to see whether it qualified, and the V3-V4 region (468 bp) of 16S rDNA was selected for primer design (forward primer: 5′-CCTACGGGNGGCWGCAG-3′; reverse primer: 5′-GACTACHVGGGTATCTAATCC-3′), followed by PCR amplification of the DNA. The amplified products needed to be further verified via 2% agarose gel electrophoresis, and a DNA library was constructed. Qualified DNA libraries were sequenced using an Illumina sequencer and obtained for a bioinformatics analysis.

After the samples were sequenced, raw data were obtained, paired-end sequences were assigned for overlapping splicing, and then the raw data were quality filtered to obtain high-quality data and remove redundant chimeras. Dereplication was carried out using DADA2 to obtain feature tables and feature sequences, and then a diversity analysis and species classification annotation were carried out. Alpha diversity and beta diversity were calculated via normalization to the same sequences randomly. Most bacteria were identified and analyzed based on the sequencing results.

### 2.9. Serum-Related Metabolites Extraction and Non-Targeted Metabolomics Analysis

The collected serum samples were thawed on ice, and metabolites were extracted with 50% methanol buffer. Briefly, 20 μL of sample was extracted with 120 μL of precooled 50% methanol, vortexed for 1 min, and incubated at room temperature for 10 min; the extraction mixture was then stored overnight at −20 °C. After centrifugation at 4000× *g* for 20 min, the supernatants were transferred into new 96-well plates. The samples were stored at −80 °C prior to an LC-MS analysis. In addition, pooled QC samples were also prepared by combining 10 μL of each extraction mixture. 

All samples were acquired by the LC-MS system by following the machine’s orders. Firstly, all chromatographic separations were performed using a Vanquish Flex UHPLC system (Thermo Fisher Scientific, Bremen, Germany). An ACQUITY UPLC T3 column (100 mm × 2.1 mm, 1.8 µm, Waters, Milford, CT, USA) was used for the reversed-phase separation. The column oven was maintained at 35 °C. The flow rate was 0.4 mL/min, and the mobile phase consisted of solvent A (water, 0.1% formic acid) and solvent B (acetonitrile, 0.1% formic acid). Gradient elution conditions were set as follows: 0~0.5 min, 5% B; 0.5~7 min, 5% to 100% B; 7~8 min, 100% B; 8~8.1 min, 100% to 5% B; 8.1~10 min, 5% B.

A high-resolution tandem mass spectrometer, Q-Exactive (Thermo Scientific, Waltham, MA, USA), was used to detect metabolites eluted from the column. The Q-Exactive was operated in both positive and negative ion modes. Precursor spectra (70–1050 *m*/*z*) were collected at 70,000 resolution to hit an AGC target of 3 × 10^6^. The maximum injection time was set to 100 ms. The top 3 configurations to acquire data were set in the DDA mode. Fragment spectra were collected at 17,500 resolution to hit an AGC target of 1 × 10^5^ with a maximum injection time of 80 ms. In order to evaluate the stability of the LC-MS during the whole acquisition, a quality control sample (pool of all samples) was acquired after every 10 samples.

### 2.10. Integrated Microbiome–Metabolome Analysis

A Spearman correlation analysis was performed to determine the relationships between the differential flora, between the differential metabolites, and between the differential metabolites and differential flora. This was achieved by performing a Spearman correlation analysis on the differential secondary metabolites obtained from the metabolomics screening and on the flora with significantly differential genus levels obtained from the 16S rDNA sequencing analysis. Based on the calculation results, suitable screening conditions were selected to determine the final correlations between the differential flora and differential metabolites and network diagrams, etc.

### 2.11. Statistical Analysis

All data were initially sorted in Excel 2021, and data from the fermentation trial were analyzed using one-way ANOVA through Design-expert12.0. Subsequent data were analyzed using SPSS (version 20.0), and differences between the control and experimental groups were analyzed using the *t*-test. The results are expressed as the mean ± standard deviation (SD), and *p* < 0.05 was considered to indicate a statistically significant difference. 

## 3. Results

### 3.1. Determination of the Growth Curve of Candida utilis Feed

Figure 1A shows the growth curve of Candida utilis drawn according to the OD value of the strain liquid. It can be seen in the figure that the strain concentration increased moderately, and the strain growth was slower. Furthermore, 0–8 h was the lag growth phase; 8–16 h was the logarithmic growth phase, where the strain rapidly grew to the highest point and showed the fastest reproduction; and 16–24 h was the stationary phase, where the strain concentration changed in a more stable manner. Thus, 20 h was chosen as the subsequent strain fermentation time.

### 3.2. Single-Factor Experiment Results of Fermentation Medium Components for Candida utilis Feed

It can be seen in Figure 1B that, when the yeast extract concentration was 2% and KH_2_PO_4_ concentration was 0.2%, the OD value and number of yeast cells of the yeast protein feed increased gradually with the concentration of the glucose master liquor. The OD value and number of yeast cells reached the maximum value of 0.831 and 3.32 × 10^8^/mL, respectively, when 8% glucose master liquor was added. 

In Figure 1C, it can be seen that, when the concentration of the glucose master liquor was 8% and the concentration of KH_2_PO_4_ was 0.2%, with the gradual increase in the corn steep liquor concentration, there was no significant change in the OD value; additionally, the number of yeast cells showed a decreasing trend in the fermentation liquid of *Candida utilis* protein feed. When 1% corn steep liquor was added, the OD value and number of yeast cells of the fermentation liquor reached the maximum value of 0.704 and 2.48 × 10^8^ cells/mL, respectively.

It can be seen in Figure 1D that, when the concentration of the glucose master liquor was 8% and the concentration of the corn steep liquor was 1%, the OD value and number of yeast cells of the fermentation liquid of *Candida utilis* protein feed gradually increased with the concentration of the inorganic salt KH_2_PO_4_; additionally, the OD value and number of yeast cells reached the maximum value of 0.697 and 2.33 × 10^8^ cells/mL, respectively, when 0.15% KH_2_PO_4_ was added. 

The concentrations of 8% glucose master liquor, 1% corn steep liquor, and 0.15% KH_2_PO_4_ were chosen for the subsequent experiment.

### 3.3. RSM Results of Culture Medium Components of Candida utilis Feed

#### 3.3.1. Optimization of Medium Components

Based on the results of the single-factor test, a Box–Behnken central combination design was used in the response surface analysis to select the three factors of glucose master liquor concentration, corn steep liquor concentration, and inorganic salt KH_2_PO_4_ concentration and the three levels of the three factors for the optimization test. Furthermore, the OD value and number of yeast cells were selected as the response values for the multi-factor interaction test, and then the data were analyzed using Design-Expert 12.0 software. The levels of each factor are shown in Table 2, the experiment groups and results are shown in Table 3, and the ANOVA results are shown in Table 4 and Table 5.

A regression analysis was performed on the data in Table 3 using Design-Expert 12.0 software to obtain a quadratic regression equation for determining the OD values and number of yeast cells in response to the three factors:OD_600_ = 0.6760 + 0.0136A + 0.0144B − 0.0025C + 0.0062AB − 0.0140AC − 0.0090BC − 0.0396A^2^ − 0.0196B^2^ + 0.0006C^2^
Number of yeast cells = 2.77 + 0.0800A + 0.0475B + 0.0675C − 0.0150AB − 0.1000AC − 0.0050BC − 0.2930A^2^ − 0.0980B^2^ − 0.1280C^2^

In Table 4, the *F*-value was 40.02 and the *p*-value was < 0.0001, indicating that the model difference was highly significant, and the F-value in the out-of-fit term was 0.9886 and the *p*-value was = 0.9946 > (0.05), indicating that the difference was not significant enough for an optimal analysis. The model correlation coefficient of r^2^ = 0.9809 indicated that the regression equation could better fit the variation in OD in the fermentation liquid, and 98.09% of the variation in OD came from the selected factors. The correction coefficient of determination of r^2^_adj_ = 0.9564 indicated that the actual value of OD was close to the predicted value of the regression equation, indicating that the OD value could be predicted and analyzed well using this equation. The coefficient of variation (CV) was 0.9295 and the signal-to-noise ratio (adequate precision) was 19.0111, further indicating that the model is accurate and credible and can be used for the study of the culture medium process of protein feeds containing *Candida utilis*. The order of the effect of each factor on the response value was B > A > C. The effect of A, B, AC, A^2^, and B^2^ on the OD value in the equation was highly significant (*p* < 0.01); the effect of BC on the OD value was significant (*p* < 0.05); and the effect of C, AB, and C^2^ on the OD value was not significant (*p* > 0.05).

In Table 5, the *F*-value was 31.14 and the *p*-value was < 0.0001, indicating that the model difference was highly significant, and the F-value in the out-of-fit term was 0.0594 and the *p*-value was = 0.9876 > (0.05), indicating that the difference was not significant enough for an optimal analysis. The model correlation coefficient of r^2^ = 0.9756 indicated that the regression equation could better fit the variation in the number of yeast cells, and 97.56% of the variation in the number of yeast cells came from the selected factors. The correction coefficient of determination of r^2^_adj_ = 0.9443 indicated that the actual value of the number of yeast cells was close to the predicted value of the regression equation, indicating that the number of yeast cells could be predicted and analyzed well using this equation. The coefficient of variation (CV) was 1.92 and the signal-to-noise ratio (adequate precision) was 17.9965, further indicating that the model is accurate and credible and can be used for the study of the culture medium process of *Candida utilis* feed. The order of the effect of each factor on the response value was A > C > B. The effects of AC, A^2^, B^2^, and C^2^ on number of yeast cells in the equation were highly significant (*p* < 0.01), and the effects of AB and BC on the yeast cell number were not significant (*p* > 0.05).

#### 3.3.2. Interaction Analysis of Factors

To further study the effects of the glucose master liquor, corn steep liquor, and KH_2_PO_4_ concentrations on the OD value and number of yeast cells, 3D plots, and contour plots of the interactions among the factors were drawn based on the model regression equation. If the slope in the 3D plot is steeper, then it means that the interaction between the two factors on the horizontal coordinate is significant and has a greater effect on the OD value and number of yeast cells; if the slope in the 3D plot is milder, then it means that the interaction between the two factors on the horizontal coordinate is not significant and has little effect on the OD value and number of yeast cells. Furthermore, the contour plot is a flat projection of a 3D plot, and it can reflect the significance of the interactions between factors more intuitively. If the shape of contour plot is closer to an ellipse, then it means that the interaction between the two factors is significant and has a greater effect on the OD value and number of yeast cells; if the shape of contour plot is closer to a circle, then it means that the interaction between the two factors is not significant and has little effect on the OD value and number of yeast cells.

In Figure 2C–F, the steepness of the slope of the 3D plot is obvious and the contour plot is nearly elliptical, indicating that the interaction of glucose master liquor concentration and KH_2_PO_4_ concentration and that of the corn steep liquor concentration and KH_2_PO_4_ concentration are significant and have a large effect on the OD value. In Figure 3C,D, the steepness of the slope of the 3D plot is obvious and the contour plot is nearly elliptical, indicating that the interaction between the glucose master liquor concentration and KH_2_PO_4_ concentration is significant and has a greater effect on the number of yeast cells. The above results were consistent with the model regression ANOVA results.

The quadratic regression equation and response surface methodology of variance predicted the optimal culture medium components as follows: a glucose master liquor concentration of 8.3%, a corn steep liquor concentration of 1.2%, and a KH_2_PO_4_ concentration of 0.14% The model predicted an OD value of 0.681 and the number of yeast cells to be 2.77 × 10^8^/mL. The above optimal optimization scheme was tested and verified, and the results of three parallel tests were averaged, resulting in an OD value of 0.670 and a yeast cell count of 2.72 × 10^8^/mL, which were less different from the predicted values, so the model can be used for the optimization of culture medium components of *Candida utilis* feed.

### 3.4. Growth Performance

The results of the growth performance of Dongliao black pigs are shown in Table 6. Significant differences in the results of average daily gain (ADG), average daily feed intake (ADFI), and feed conversion ratio (FCR) were not observed.

### 3.5. Serum Biochemical Variables

In order to evaluate the effect of Candia utilis on piglet health, we measured a number of biochemical indicators in serum. The results of the serum biochemical variables are shown in Table 7. Significant differences in all serum biochemical variables were not observed. However, the addition of 2.5% CU had a tendency to reduce the ALT, AST, TC, TG, and LDL levels in pig serum compared with CON treatment.

### 3.6. Effect of Candida utilis onFecal Microflora

In this study, we used 16S rDNA sequences to detect the profiles of fecal microbes. As shown in Figure 4A, we obtained a total of 3384 ASVs from the two groups: 1350 and 992 were unique to each of the two groups, and 1042 were common in the two groups. The LEfSe analysis showed statistical differences in the identified microbiota between the treatments (B). The results showed that *c_Clostridia*, *o_Oscillospirales*, and *f_Prevotellaceae* were the marker microorganisms of the CON group. *c_Bacilli*, *f_Lactobacillaceae*, and *s_Clostridium_sensu_stricto_1_unclassified* were the marker microorganisms of the 2.5% CU group.

At the phylum level, 26 different phyla were detected in fecal contents between the CON group and the 2.5% CU group. In the CON group, *Firmicutes* (69.77%), *Bacteroidetes* (21.14%), *Actinobacteria* (4.01%), and *Proteobacteria* (3.36%) were the dominant phyla. In the 2.5% CU group, *Firmicutes* (71.65%), *Bacteroidetes* (18.07%), *Actinobacteria* (6.58%), and *Proteobacteria* (2.20%) were the dominant phyla. The relative abundance of *Firmicutes* and *Bacteroidetes* in the CON group and 2.5% CU group was 90.91% and 89.72%, respectively. Compared with the CON group, the relative abundance of *Firmicutes* increased by 1.88% in the 2.5% CU group and the relative abundance of the *Bacteroidetes* increased by 3.07%, with no significant difference (*p* > 0.05). Compared with the CON group, the relative abundance of *Actinobacteria* in the fecal microbiota increased by 2.57% with the addition of 2.5% *Candida utilis*, with a significant difference (*p* < 0.05). The relative abundance of *Proteobacteria* decreased by 1.16%, but without a significant difference (*p* > 0.05).

At the genus level, Figure 5 compares the relative abundances of the 30 most abundant genera. The five dominant genera with the highest relative abundances between the CON group and 2.5% CU group were *Clostridium_sensu_stricto_1*, *Lactobacillus*, *Muribaculaceae_unclassified*, *Prevotella_9*, and *Faecalibacterium.* Compared with the CON group, *Clostridium_sensu_stricto_1*, *Lactobacillus*, and *Muribaculaceae_unclassified* were significantly higher in the experimental group, and *Faecalibacterium* was significantly lower in the experimental group (*p* < 0.05).

### 3.7. Supplementation with Candida utilis in a Basic Diet Is a Discriminating Factor of the Plasma Metabolome

In order to investigate the extent to which *Candida utilis* could separate the plasma metabolome between the CON group and the 2.5% CU group, we performed an untargeted metabolomics analysis on the plasma samples (Figure 6). PCA was applied to investigate the clustering trends of the metabolome between the CON group and the 2.5% CU group, and to exclude possible outliers. As shown in Figure 6A, the QC samples, which are blue dots, are well aggregated, indicating that the test instruments are stable. The CON group was not well aggregated compared with the 2.5% CU group, indicating that there was no significant difference between the 2.5% CU group and CON group.

A total of 227 metabolites were detected in the untargeted metabolome analysis. The two sides of the two perpendicular threshold lines formed after taking log2 for the *p*-value of the X-axis are the differential metabolites, whereas the parts above the parallel threshold line formed after taking -log10 for the *p*-value of the Y-axis are the differential metabolites. Combining the above two points, as shown in Figure 6B, compared with the CON group, the green dots are the downregulated metabolites, and the red dots are the upregulated metabolites.

A correlation heat map was made by selecting 12 of these metabolites with relative differences. As shown in Figure 6C, compared with the CON group, Cyclo−prolylglycine−M155T143, Imidazoleacetic acid−pos−M127T146, Telbivudine−pos−M243T146, Kynurenic acid−neg−M207T147, Taurocholate−pos−M517T195, Theobromine−neg−M181Y174, and O−Demethylfonsecin−pos−M181T174 were upregulated and 2−Melthyl−4,5−benzoxazole−neg−M134T171, 6−Hydroxycoumarin−pos−M180T171, Rimantadine−pos−M180T207, 3,5−Dimethoxycinnamic acid−pos−M209T212, and Rubiadin−pos−M255T194 were downregulated. Figure 6D shows the metabolic pathways that the 12 differential metabolites were enriched in, as identified in a KEGG analysis, including tryptophan metabolism, primary bile acid metabolism, histidine metabolism, taurine and hypotaurine metabolism, and caffeine metabolism.

### 3.8. Fecal Microbiota Correlated with the Plasma Metabolome in Candida utilis

After performing the fecal flora assay and the non-targeted metabolomic analysis, we found some correlations between the results. Thus, we hypothesized that there is a correlation between these two metabolisms. As shown in Figure 7A, we detected a total of 10 metabolites associated with 27 genera, of which 5 metabolites were more strongly correlated: Cyclo−prolylglycine, Imidazoleacetic acid, Telbivudine, Theobromine, and 6−Hydroxycoumarin. As shown in Figure 7B, the network results were analyzed based on the correlation between the metabolites and genera, and it can be seen that the genera associated with these five metabolites are denser. For example, is *g_Lactobacillus*, *g_HT002*, *g_Campylobacter*, *g_Intestinimonas*, *g_Selenomonadaceae_unclassified*, *g_Peptococcus*, *g_Ruminococcus]_torques_group*, *g_UBA1819*, *g_Subdoligranulum*, *g_Erysipelotrichaceae_UCG−006* and *g_Anaerofustis* showed positive correlations with Cyclo−prolylglycine, and *g_Atopostipes*, *g_Ignatzschineria*, *g_Fibrobacter*, *g_Novispirillum*, *g_Prevotellaceae_unclassified*, *g_Coprococcus*, *g_Prevotella*, *g_Eubacterium]_nodatum_group*, *g_Lysinibacillus*, *g__UCG−005*, *g_Eubacterium]_xylanophilum_group*, *g_Defluviitaleaceae_UCG−011*, and *g_Bacteroidales_RF16_group_unclassified* showed negative correlations with Cyclo−prolylglycine.

## 4. Discussion

Yeast can be added to animal feed as an inexpensive, safe, and novel protein source, as it contains a high protein content and well-balanced amino acids. *Candida utilis* feed is produced from different agricultural, forestry, and industrial wastes, so its use contributes to eliminating pollutants and the recycling of materials from waste [18,19,20]. The glucose master liquor used in this experiment is the liquid left over after the production of crystalline glucose, which can become a source of pollution if not fully utilized. Corn steep liquor is a byproduct of corn starch production using the wet process. Concentrated corn steep liquor has a mushy aromatic odor, high nutritional value, and low price; promotes microbial growth and metabolism; and is a high-quality nitrogen source in process production. Although the current application of these two raw materials as the main components of fermented protein feeds is relatively limited, they are expected to be widely used based on the application results.

During fermentation, the medium component has a great influence on the growth and reproduction of yeast cells. We firstly optimized the medium component and the single factor results showed that 8% glucose master liquor, 1% corn steep liquor, and 0.15% KH_2_PO_4_ could meet the growth demand of the organism, and more yeast cells increased the synthesis rate of fermentation products. Although the concentration of the carbon source needs to be sufficient to supply the growth and reproduction of *Candida utilis*, a too-high concentration will inhibit the uptake of other growth factors by *Candida utilis* and slow down the metabolism rate of the bacterium. Zhang et al. [21] used glucose and other substances in combination with *Candida utilis* and *Saccharomyces cerevisiae* to produce a single-cell protein with a protein content of 43.59% and rich in various amino acids. Zhao et al. [22] used waste capsicum powder as substrate and 3% corn steep liquor as a nitrogen source to ferment feed using *Candida utilis*, and this improved the growth rate of the yeast and increased the protein content. Athar et al. [23] also applied *Candida utilis* as a fermenting bacterium to produce feed additives for animal breeding with beet pulp as a substrate and 1% corn steep liquor as a nitrogen source. When yeast cells lack the supply of phosphate and metal elements, the reaction of nutrient uptake stops, leading to the inhibition of fermentation [24]. If the concentration of inorganic salt KH_2_PO_4_ is continuously increased, then it may affect the cellular metabolic function of the bacterium and its internal and external osmotic pressures may become too high, resulting in a decrease in the OD value and number of yeast cells [25]. Favorable results were obtained by optimizing the medium components through response surface methodology, which could be applied in actual production, proving that these raw materials are beneficial to the growth and reproduction of yeast and achieve a considerable protein content in the final fermentation products.

At present, *Candida utilis* feed is popularly used in the aquaculture industry, and there are few studies in livestock and poultry farming. Moreover, feeding with *Candida utilis* feed could improve the growth performance and nutrient digestibility of aquatic animals, and it could also alleviate the intestinal inflammation caused by soybean meal [26,27,28]. Cruz et al. [5] studied the effects of *Candida utilis* added to the diets of weaned piglets and found that the addition of yeast to diets did not affect growth performance compared with the control. In this study, compared with the CON group, the addition of 2.5% *Candida utilis* protein feed to the basic diet caused no significant differences in the average daily gain (ADG), average daily feed intake (ADFI), or F/G of black piglets. Similar results have been reported in cattle [29]. However, other studies proved that *Candida utilis* not only improves the growth performance of weaned piglets but also reduces diarrhea [30,31]. These different results may be due to factors such as yeast composition, addition amount, animal category, and feeding environment, but what can be demonstrated is that the use of *Candida utilis* as a source of protein has no adverse effects on growth performance in livestock farming and that growth requirements can be maintained.

Serum biochemical indicators refer to the levels of various chemical components in the plasma, including the levels of many metabolites of the body. Serum is similar to the interstitial fluid of the body’s tissues, and it is more accurate in indicating the physiological situation of the body and more sensitive in indicating pathological changes. For example, ALT and AST are the most widely used biochemical indicators of hepatocellular injury in clinical practice. A previous study showed that the addition of yeast extract significantly reduced the AST and ALT levels in the blood of pikeperch without harming their liver, intestinal, or kidney functions [32]. In this experiment, the addition of 2.5% CU resulted in a decreasing in AST and ALT in the weaned piglets, indicating that the addition of CU did not damage their liver function, but rather maintained their healthy liver function, improved their immunity and facilitated their growth. As TC, TG, LDL-C, and HDL-C are important indicators in blood lipid examinations, they are often used to assess atherosclerosis, or cardiovascular disease, etc. Azizi et al. [33] found that addition of yeast reduced the TG content in broiler plasma. In this experiment, the addition of *Candida utilis* had a tendency to reduce this indicator compared with the control group. This suggests that *Candida utilis*, as a novel protein source, has the effect of improving blood lipids, and it has been reported that the addition of probiotics to the diet can inhibit lipid biosynthesis and increase fatty acid catabolism [34].

The intestinal flora of mammals is diverse and increases with the stage of growth, and, during the weaning of piglets, physiological stress can lead to changes in fecal flora [35]. Therefore, we assessed microbiota diversity via 16S rDNA sequencing of fecal flora. The results of this experiment showed that *Firmicutes*, *Bacteroidota*, *Actinobacteriota*, and *Proteobacteria* were the most dominant phyla in the piglets. This is consistent with the results of [36]. In addition, the relative abundance of *Actinobacteriota* increased significantly compared with the control after the addition of yeast at the phylum level. *Actinobacteriota* is the dominant flora in the animal intestinal tract, and its metabolism produces bioactive substances with anticancer and antibacterial effects [37]. The increase in the abundance of *Actinobacteriota* with the addition of yeast protein feed may alleviate weaning stress and improve the intestinal immunity of piglets, but it is likely that the whole experimental period was too short and did not fully reflect the better growth performance. At the genus level, the current results revealed a high relative abundance of *Clostridium_sensu_stricto_1*, *Lactobacillus*, and *Muribaculaceae_unclassified* in the 2.5% CU group compared with the control group. Yu et al. [38] added *Bacillus licheniformis* to piglet diets, which also increased the relative abundance of *Clostridium_sensu_stricto_1* and *Lactobacillus*. *Clostridium_sensu_stricto_1* and *Lactobacillus* are members of *Firmicutes*, most of which are beneficial intestinal bacteria. *Clostridium_sensu_stricto_1* was reported to promote the adhesion of the intestinal mucus barrier to pathogens, and an increasing in its relative abundance in cecum content adversely correlated with levels of inflammatory factors [39,40]. The isolates of *Lactobacillus* have been found to be able to inhibit pathogen invasion of intestinal epithelial cells and reduce the adhesion of some pathogenic bacteria to the intestinal mucosa [41]. *Muribaculaceae* used undigested food to produce short-chain fatty acids, such as propionate and butyrate. They play an important role in regulating gut health in animals [42]. We also observed an increase in *Collinsella* abundance, which was positively correlated with blood T-cell responses and may be associated with enhancing serum immunity in weaned piglets [43]. In addition, the relative abundance of *Prevotella 9* showed a decreasing trend in the 2.5% CU group, which is consistent with Iakhno et al. [44]. High levels of *Prevotella* not only exacerbate inflammation but are also associated with obesity and high blood pressure [45]. However, some other different results were found, such as an increasing in the abundance of *Faecalibacterium*; thus, further research is needed to investigate the reasons for this. However, it has been confirmed that *Candida utilis* feed can regulate the intestinal flora of weaned piglets, thereby increasing the content of beneficial bacterial genera and protecting the intestinal barrier by inhibiting the invasion of pathogenic bacteria and inflammation; this is conducive to reducing the weaning stress of piglets, thus guaranteeing their healthy growth. Furthermore, the abundance of *Clostridium_sensu_stricto_1* in finishing pigs decreased significantly with decreasing protein levels [46]. Thus, these results could indicate that it is feasible to apply protein feed fermented by *Candida utilis* as a novel protein source by virtue of its rich protein content.

In order to explore the effect of *Candida utilis* feed on the metabolic levels of the weaned piglets, metabolomics was used to provide a more comprehensive view of the changes in their plasma metabolites. In this experiment, as determined from our metabolomics results, the metabolic pathways enriched by the 12 differential metabolites were tryptophan metabolism, primary bile acid metabolism, histidine metabolism, taurine and hypotaurine metabolism, and caffeine metabolism. Histidine is a semi-essential amino acid, and it is metabolized mainly by gut microbes. Li et al. [47] found that feeding a probiotic complex to pigs may affect the metabolism of histidine by producing histidine, which was beneficial in improving the digestive function of the pig intestine. Another study revealed the important role of histidine in the treatment of inflammatory responses, oxidative stress, and metabolic disorders [48]. Tryptophan is an essential amino acid that plays an important role in regulating appetite and the stress response, maintaining the immune system and protecting the health of the body [49]. Primary bile acid metabolism is involved in glycolipid metabolism, facilitates lipid absorption, regulates intestinal flora, reduces inflammation, and maintains the integrity of the intestinal epithelial barrier [50]. Hence, it can be concluded that, in weaned piglets, the addition of yeast protein feed can regulate metabolism levels and will lead to a higher content of metabolites related to digestion and immune function. This has a positive regulatory effect on the regulation of intestinal flora, maintenance of intestinal barriers, reduction in the occurrence of diseases, and improvement of the immune system, maintaining the normal growth and development of piglets. In this study, we performed a combined analysis of fecal microbiota 16S rDNA sequencing and untargeted metabolomics and found close associations between the gut flora and in vivo metabolism of weaned piglets fed the *Candia utilis* protein feed. A total of 10 plasma metabolites were correlated with 27 genera of bacteria as revealed by the results of the combined multi-omics analysis. Metabolites such as Cyclo-prolylglycine, Telbivudineand Theobromine 6Hydroxycoumarin showed highly significant positive correlations with *Lactobacillus*, *Anaerofustis*, and *Atopostipes*, which are beneficial in *Firmicutes*. A study found that Cyclo-prolylglycine exhibited neuroprotective potential in rats with glutamate neurotoxicity [51]. Telbivudine was used in clinical applications to treat chronic hepatitis B and showed no adverse effects and Theobromine could reduce diet-induced obesity through fat modulation [52,53]. 6-Hydroxycoumarin is a natural compound with an aromatic odor widely distributed in Umbelliferae, and it has high clinical value as an anticancer, antifungal, and anti-inflammatory agent [54]. Therefore, we concluded that the addition of *Candida utilis* protein feed can regulate the intestinal flora of piglets, act as an anti-inflammatory, and maintain intestinal homeostasis through the metabolic pathways affected by these metabolites, such as primary bile acid metabolism, histidine metabolism, and tryptophan metabolism.

## 5. Conclusions

In this study, utilizing response surface methodology, we first optimized the medium composition of feed fermented by *Candida utilis*, using glucose master liquor and corn steep liquor as substrates. Second, the effects of the *Candida utilis* protein feed on the growth performance, intestinal flora, and metabolic levels of weaned piglets were explored using 16S rDNA sequencing combined with metabolomics. The addition of *Candida utilis* feed to the diets had positive effects on performance and serum immunity, altered the composition of metabolites and microbial communities and regulated the gut microbiota of the weaned pigs. The results of this study provide a theoretical basis for the application of *Candida utilis* in pig feed.

## Figures and Tables

**Figure 1 animals-14-00306-f001:**
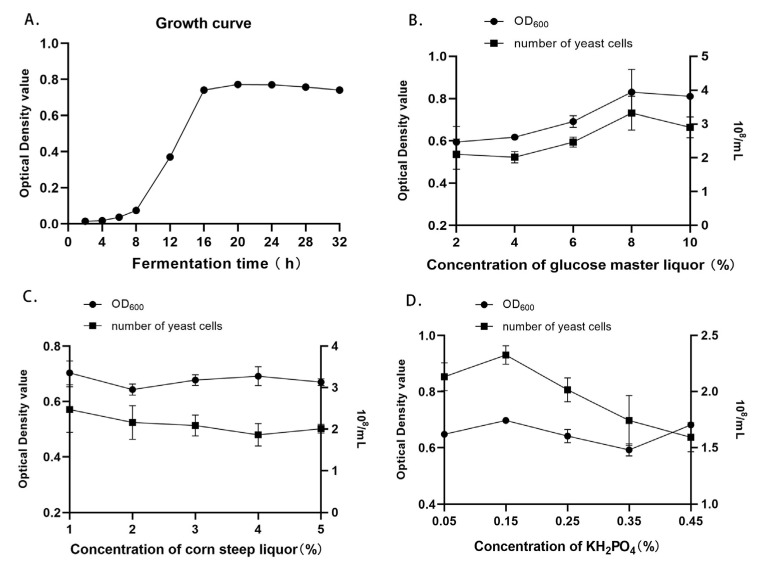
The growth curve of *Candida utilis* feed (**A**). Effect of concentration of glucose mother liquor, corn steep liquor and inorganic salt KH_2_PO_4_ on OD value and number of yeast cells in fermentation liquid (**B**–**D**).

**Figure 2 animals-14-00306-f002:**
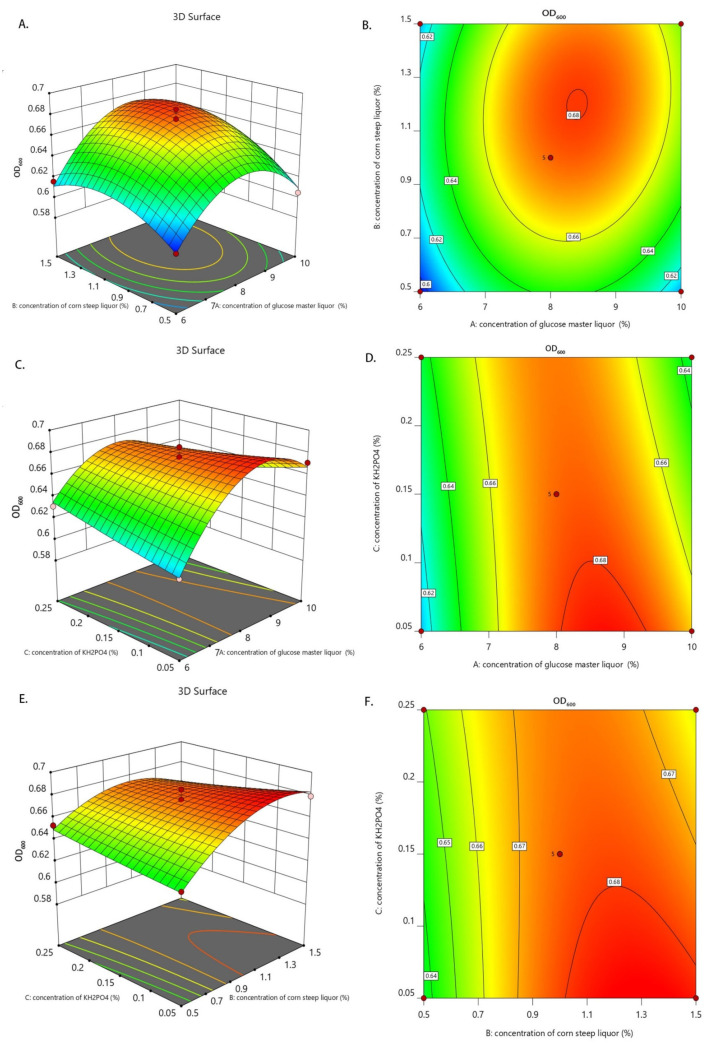
Response surface of fermentation culture components on OD value. Three-dimensional plot (**A**) and contour (**B**) of concentration of glucose master liquor and concentration of corn steep liquor; 3D plot (**C**) and contour (**D**) of concentration of glucose master liquor and concentration of KH_2_PO_4_; 3D plot (**E**) and contour (**F**) of concentration of corn steep liquor and concentration of KH_2_PO_4_. In the 3D plot, if the opening of the face is downward, then the response value has extreme values; if the opening of the face is upward, then the response value has minimum values. The more curved the line is, the greater the influence of the study factors on the results, and the faster the color change indicates a greater slope—here, there is a more significant influence on the results. The contour plots show the projection of 3D at the bottom.

**Figure 3 animals-14-00306-f003:**
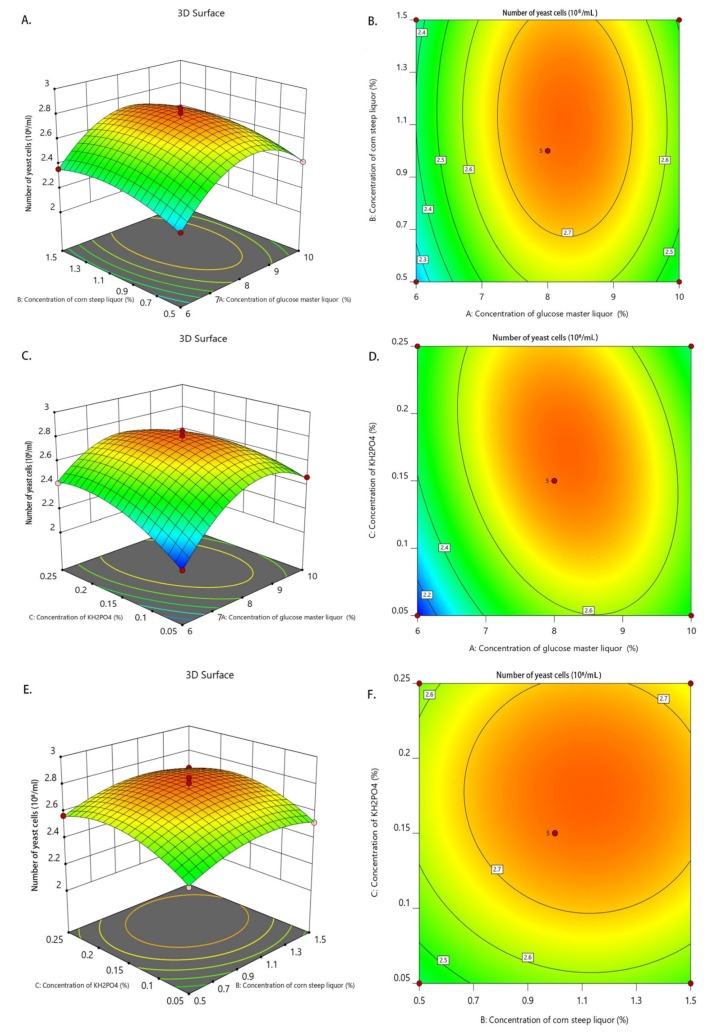
Response surface of fermentation culture components on number of yeast cells. Three-dimensional plot (**A**) and contour (**B**) of concentration of glucose master liquor and concentration of corn steep liquor; 3D plot (**C**) and contour (**D**) of concentration of glucose master liquor and concentration of KH_2_PO_4_; 3D plot (**E**) and contour (**F**) of concentration of corn steep liquor and concentration of KH_2_PO_4_. In the 3D plot, if the opening of the face is downward, then the response value has extreme value; if the opening of the face is upward, then the response value has minimum value. The more curved the line is, the greater the influence of the study factors on the results, and the faster the color change indicates a greater slope—here, there is a more significant influence on the results. The contour plots show the projection of 3D at the bottom.

**Figure 4 animals-14-00306-f004:**
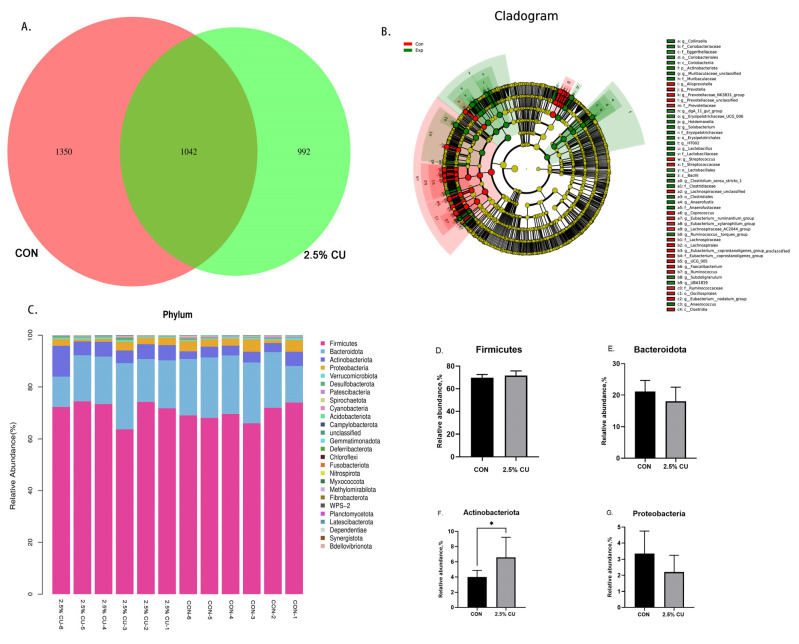
Fecal microbiota regulation by *Candia utilis* at the phylum level. (**A**) Venn diagram of shared and specific ASVs among groups. (**B**) The LEfSe analysis (LDA score ≥ 3) identified the biomarker species. The different circle layers radiate from the inside to the outside represents the seven taxonomic levels of phylum, order, family, genus, and species, respectively. The diameter of the dots represents species abundance. Yellow color indicates that the species is not significantly different in the groups, and red color indicates that the species is significantly different in the groups. (**C**) Relative abundance of fecal microbial phyla. (**D**) Relative abundance of *Firmicutes*. (**E**) Relative abundance of *Bacteroidota*. (**F**) Relative abundance of *Actinobacteriota*. (**G**) Relative abundance of *Proteobacteria.* Data are expressed as the means ± SDs, * *p* < 0.05.

**Figure 5 animals-14-00306-f005:**
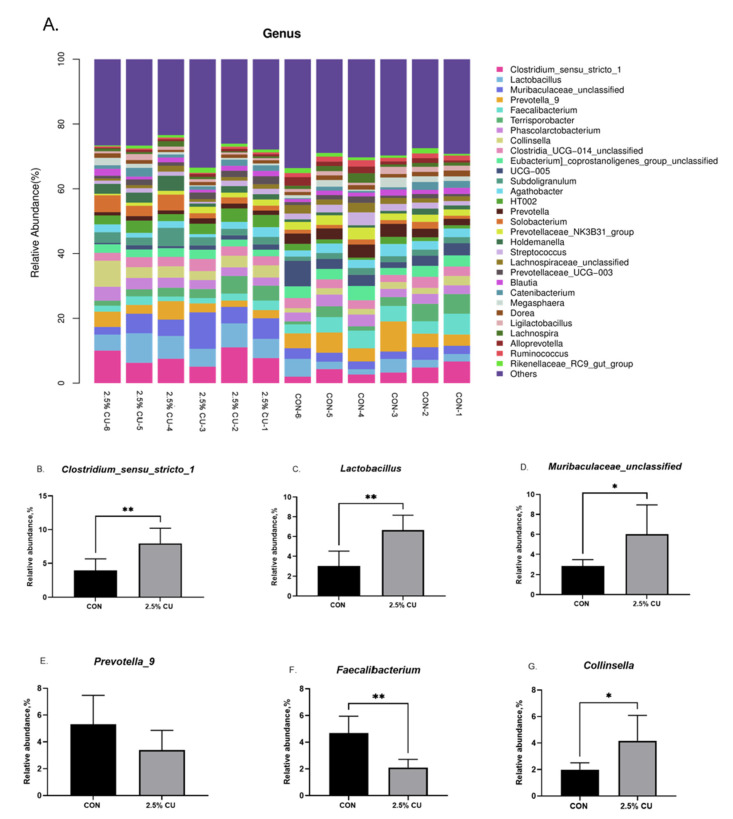
Gut microbiota regulation by *Candida utilis* at the genus level. (**A**) Relative abundance of fecal microbial genera. The relative abundances of the microbial genera. *Clostridium_sensu_stricto_1* (**B**), *Lactobacillus* (**C**), *Muribaculaceae_unclassified* (**D**), *Prevotella_9* (**E**), *Faecalibacterium* (**F**), *Collinsella* (**G**). Data are expressed as the means ± SDs, * *p* < 0.05, ** *p* < 0.01.

**Figure 6 animals-14-00306-f006:**
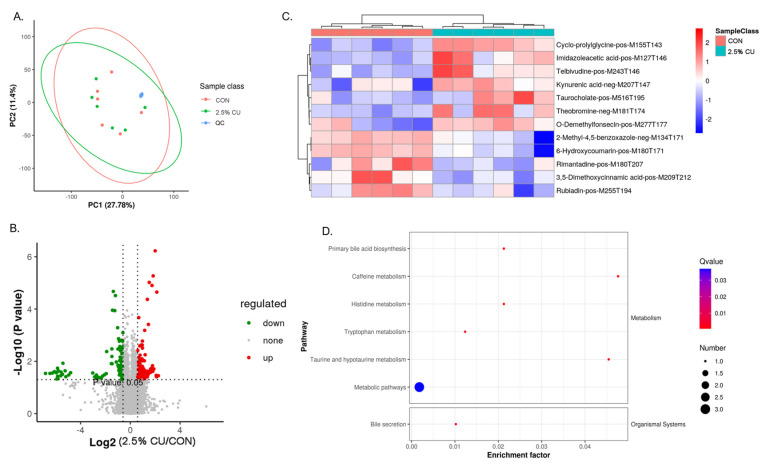
Variance analysis of plasma metabolite concentrations from supplementing the *Candida utilis* in the basic diet. (**A**) PCA analysis of metabolites. (**B**) Volcano plot of differential metabolites. Upregulated and downregulated metabolite concentrations are given in red and blue, respectively. (**C**) Correlation heatmap of differential metabolites. (**D**) Pathways enriched to 12 differential metabolites. From red to blue, the darker the color, the higher the enrichment level.

**Figure 7 animals-14-00306-f007:**
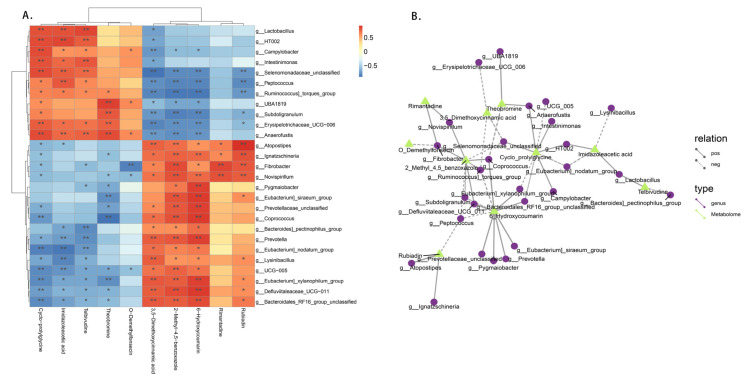
(**A**) Multi-omics association studies on the fecal microbiota and the plasma metabolome with *Candida utilis*. Red represents positive correlation, blue represents negative correlation, and the darker the color, the stronger the correlation. * represents the results of statistical test: * represents significant, ** represents highly significant. (**B**) Analysis of metabolites and the regulation of 1-gut flora network. The different nodes in the diagram represent different flora or metabolites, the shape of the flora are all circles and the shape of the metabolites are all triangles. The lines between the flora and metabolites represent the correlation between them, where the solid line represents positive correlation and the dashed line represents negative correlation.

**Table 1 animals-14-00306-t001:** Composition of experimental diets (%, as fed basis).

Ingredients, %	CON	2.5% CU
Corn	40	40
Storage japonica brown rice	10	10
Flour	6	6
Soybean meal (43.5% CP)	10	10
Fermented Soybean meal	5	5
Puffed soybean	5	5
Fish meal (65.2% CP)	3	3
Sucrose	2	2
Glucose	1	1
Whey powder	10	10
*Saccharomyces cerevisiae*	2.5	0
*Candida utilis*	0	2.5
Soybean oil	1	1
CaH_2_(PO_4_)_2_	0.6	0.6
Stone powder	0.5	0.5
NaCl	0.2	0.2
Lysine Hydrochloride (Lys) (98%)	0.63	0.63
Methionine (Met)	0.12	0.12
Threonine (Thr)	0.15	0.15
Tryptophan (Trp)	0.05	0.05
Cr_2_O_3_	0.25	0.25
Premix ^1^	2	2
Total	100	100
Nutrition level (%)		
DE (MJ/kg)	14.01	14.06
CP	18.6	18.6
Lys	1.37	1.36
Met	0.40	0.41
Thr	0.78	0.78
Trp	0.24	0.24

Note: CON, the group was supplemented the basic diet; 2.5% CU, the group was supplemented with 2.5% *Candida utilis* feed. ^1^ Premixes provided per kilogram of feed: vitamin A, 10,000 IU; vitamin D3, 2000 IU; vitamin E, 24 mg; vitamin K, 2.8 mg; vitamin B1, 1.2 mg; vitamin B2, 7.35 mg, vitamin B6, 1.14 mg; nicotinamide, 35 mg; pantothenic acid, 19.85 mg; Mn, 30 mg; Zn, 100 mg; Fe, 90 mg; Cu, 80 mg; Se, 0.2 mg.

**Table 2 animals-14-00306-t002:** Factors and levels of fermentation medium components for Box–Behnken design.

Factors	Levels		
	−1	0	1
Concentration of glucose master liquor (%)	6	8	10
Concentration of corn steep liquor (%)	0.5	1.0	1.5
Concentration of KH_2_PO_4_ (%)	0.05	0.15	0.25

**Table 3 animals-14-00306-t003:** Box–Behnken experimental design and results.

Runs	Concentration of Glucose Master Liquor (%)	Concentration of Corn Steep Liquor (%)	Concentration of KH_2_PO_4_ (%)	Optical Density Value	Number of Yeast Cells (10^8^/mL)
1	1	−1	0	0.605	2.42
2	−1	−1	0	0.595	2.24
3	0	0	0	0.676	2.72
4	0	1	−1	0.679	2.52
5	0	−1	−1	0.637	2.41
6	0	0	0	0.685	2.81
7	0	0	0	0.668	2.7
8	0	0	0	0.676	2.85
9	0	−1	1	0.653	2.57
10	−1	1	0	0.616	2.36
11	−1	0	−1	0.611	2.1
12	1	1	0	0.651	2.48
13	1	0	−1	0.671	2.47
14	0	1	1	0.659	2.66
15	0	0	0	0.675	2.75
16	−1	0	1	0.631	2.42
17	1	0	1	0.635	2.39

**Table 4 animals-14-00306-t004:** Results of factorial regression model with OD value.

Source	Sum of Squares	df	Mean Square	*F*-Value	*p*-Value	Significance
Model	0.0131	9	0.0015	40.02	<0.0001	***
A—Concentration of glucose master liquor	0.0015	1	0.0015	40.89	0.0004	***
B—Concentration of corn steep liquor	0.0017	1	0.0017	45.51	0.0003	***
C—Concentration of KH_2_PO_4_	0.0000	1	0.0000	1.38	0.2791	ns
AB	0.0002	1	0.0002	4.30	0.0768	ns
AC	0.0008	1	0.0008	21.59	0.0024	**
BC	0.0003	1	0.0003	8.92	0.0203	*
A^2^	0.0066	1	0.0066	182.02	<0.0001	***
B^2^	0.0016	1	0.0016	44.65	0.0003	***
C^2^	0.0000	1	0.0000	0.0453	0.8375	ns
Residual	0.0003	7	0.0000			
Lack of Fit	0.0001	3	0.0000	0.9886	0.4829	ns
Pure Error	0.0001	4	0.0000			
Cor Total	0.0133	16				

r^2^ = 0.9809; adjusted r^2^ = 0.9564; predicted r^2^ = 0.8530; adequate precision = 19.0111; CV% = 0.9295. ns represents *p* > 0.05; * represents *p* < 0.05; ** represents *p* < 0.01; *** represents *p* < 0.001.

**Table 5 animals-14-00306-t005:** Results of factorial regression model with number of yeast cells.

Source	Sum of Squares	df	Mean Square	*F*-Value	*p*-Value	Significance
Model	0.6574	9	0.0730	31.14	<0.0001	***
A—Concentration of glucose master liquor	0.0512	1	0.0512	21.83	0.0023	**
B—Concentration of corn steep liquor	0.0181	1	0.0181	7.69	0.0275	*
C—Concentration of KH_2_PO_4_	0.0364	1	0.0364	15.54	0.0056	**
AB	0.0009	1	0.0009	0.3837	0.5553	ns
AC	0.0400	1	0.0400	17.05	0.0044	**
BC	0.0001	1	0.0001	0.0426	0.8423	ns
A^2^	0.3615	1	0.3615	154.10	<0.0001	***
B^2^	0.0404	1	0.0404	17.24	0.0043	**
C^2^	0.0690	1	0.0690	29.41	0.0010	***
Residual	0.0164	7	0.0023			
Lack of Fit	0.0007	3	0.0002	0.0594	0.9786	ns
Pure Error	0.0157	4	0.0039			
Cor Total	0.6738	16				

r^2^ = 0.9756; adjusted r^2^ = 0.9443; predicted r^2^ = 0.9469; adequate precision = 17.9965; CV% = 1.92. ns represents *p* > 0.05; * represents *p* < 0.05; ** represents *p* < 0.01; *** represents *p* < 0.001.

**Table 6 animals-14-00306-t006:** Effect of *Candida utilis* on growth performance of black piglets.

Items	CON	2.5% CU	*p*-Value
Initial BW	6.73 ± 1.26	7.09 ± 1.30	0.11
Final BW	14.05 ± 3.16	14.33 ± 3.24	0.62
ADG, kg	0.35 ± 0.08	0.34 ± 0.07	0.77
ADFI, kg	0.56 ± 0.13	0.56 ± 0.07	0.93
FCR	1.64 ± 0.09	1.69 ± 0.24	0.71

Data represent the means ± SD values. CON, control group fed a basic diet; 2.5% CU, basic diet + 2.5% *Candida utilis*. initial BW, initial body weight; final BW, final body weight; ADFI, average daily feed intake; ADG, average daily gain; FCR, feed conversion ratio.

**Table 7 animals-14-00306-t007:** Effect of the *Candida utilis* on serum biochemical variables of the black piglets.

Items	CON	2.5% CU	*p*-Value
ALT, U/L	82.22 ± 18.94	67.50 ± 10.19	0.10
AST, U/L	91.48 ± 36.41	78.53 ± 23.68	1.00
ALP, U/L	351.10 ± 100.05	366.10 ± 97.40	0.60
GGT, U/L	48.65 ± 38.78	47.79 ± 26.49	0.96
T-BIL, μmol/L	0.58 ± 0.28	0.50 ± 0.33	0.40
D-BIL, μmol/L	0.85 ± 0.67	1.30 ± 0.32	0.29
TC, mmol/L	2.23 ± 0.50	1.90 ± 0.22	0.23
TG, mmol/L	0.23 ± 0.11	0.14 ± 0.04	0.13
HDL-C, mmol/L	0.43 ± 0.10	0.35 ± 0.07	0.21
LDL-C, mmol/L	0.61 ± 0.29	0.46 ± 0.21	0.45

Data represent the means ± SD values. CON, control group fed a basic diet; 2.5% CU, basic diet + 2.5% *Candida utilis*. HDL, high-density lipoprotein; LDL, low-density lipoprotein; AST, aspartate aminotransferase; ALT, alanine aminotransferase; ALP, alkaline phosphatase; GGT, Gamma-glutamyltransferase; T-BIL, total bilirubin; D-BIL, Direct Bilirubin; TC, total cholesterol; TG, triglycerides.

## Data Availability

The data that support the findings of this study are available from the corresponding author upon reasonable request.

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
