# Peer review of "Integrated Microbiome and Metabolomics Analysis of the Effects of Dietary Supplementation with Corn-Steep-Liquor-Derived *Candida utilis* Feed on Black Pigs"

_animals, 2024, doi:10.3390/ani14020306_

Round 1

Reviewer 1 Report

Comments and Suggestions for Authors

In the manuscript entitled “Evaluation of dietary supplementation with the Candida utilis feed produced from corn steep liquor in black pigs through integrated microbiome and metabolomics analysis”, Qi et al. aimed to investigate the effect of the feeding effect of Candida utilis feed on growth performance, intestinal flora, and metabolic level of weaned Dongliao black pigs. This is a robust manuscript, which can still be improved with a few points, so I suggest a minor revision.

1、In the Results, based on the Figure 1C., the single factor experiments on concentration of corn steep liquor on OD value and number of yeast cells showed that the range of 3-5% was better, but the levels of corn steep liquor in Box Behnken design was 0.05-0.25%. In other words, the concentration range of corn steep liquor in previous single factor experiments should be narrow down?

2、In the Results, the OD value of the three parallel tests for verifying the optimal scheme should be offered in Line 346.

3、In the Materials and methods, the data ADG, ADFI, and FCR in Line 166-170 shouldn’t be described in “2.6. Sample collection”, please added in section 2.5. And the initial BW and final BW should be provided in Table 6. 

4、How to select the 12 metabolites with relative differences in Line 446? Based on VIP value or fold change? And the 200 metabolites were in positive or negative ion modes?

5、In the Materials and methods, the “number of yeast cells was determined by counting chamber” was recommended to add a reference in Line 121.

6、In the Discussion, added references in Line 527.

7、Some minor formatting issues, please add spaces in “±” in all Tables.

8、The Latin name of genus and species of microbe should be italicized. Please italicized the name of genus of microbe in Figure 5. 

Comments on the Quality of English Language

Language is fine

Author Response

We thank editor and reviewer for providing constructive feedback. We have fully revised our manuscript. Please see the attachment.

Reviewer 2 Report

Comments and Suggestions for Authors

The paper describes an experiment utilizing glucose master liquor, corn steep liquor, and Candida utilis to produce yeast feed, optimizing its components through response surface methodology. The optimal medium was determined as 8.3% glucose master liquor, 1.2% corn steep liquor, and 0.14% KH2PO4, resulting in specific fermentation conditions with an OD value of 0.670 and a yeast cell count of 2.72×108/ml. Subsequently, the impact of Candida utilis feed on Dongliao black pigs was investigated in terms of growth performance, fecal microbiota, and plasma metabolic levels.

While the experimental setup is described adequately, the paper lacks critical details. The growth performance and serum biochemical indices showed no significant differences, but these results are presented without statistical analyses or specific values, making it challenging to assess the reliability of the findings. The alteration of fecal microbiota and plasma metabolome is discussed, but the relevance of these changes to the overall health or performance of the pigs is not clearly established.

Throughout the paper, it's crucial to ensure accurate and consistent usage of scientific names for microbes. Correctly written scientific names follow the binomial nomenclature system, where the genus is capitalized, and the species is in lowercase, both in italics or underlined. Here are examples of how to correctly write scientific names:

The authors should explicitly explain how each parameter is directly related to their hypothesis. Each chosen parameter should have a clear connection to the research question or hypothesis under investigation.

The provided data is not suitable for analysis using one-way ANOVA. Revision is necessary.

Author Response

We thank editor and reviewer for your feedback and these comments concerning our manuscript. Please see the attachment.

Reviewer 3 Report

Comments and Suggestions for Authors

Evaluation of dietary supplementation with the Candida utilis feed produced from corn steep liquor in black pigs through integrated microbiome and metabolomics analysis

Dear Authors,

The manuscript is interesting and well prepared, additionally contain information about microbiome and metabolome after using diets with 2,5% different kinds of yeast (Saccharomyces cerevisiae and Candida utilis). Several corrections in text are needed.

Below I add some suggestions helpful in this process:

Line 10

Maybe the first corresponding author and the second corresponding author? That will sounds better.

Line 11, 53 and 56

Better and safer is to write: yeasts can replace other sources of the crude protein, but partially (max. 5-6% of soybean meal) it is not possible to replace all crude protein sources during diet formulation by yeasts.

Line 55

Space needed before reference number in text of manuscript: source of nutrients [1]. And so on… to [53].

Line 158

What kind of flour? (important in case of chemical composition of diet).

Line 287 and …

In the text of manuscript is P-value, must be p-value.

Line 290 and …

The same with R2, must be r2 (coefficient from sample).

Line 351 and 353

In table title is: “Anova results of regression model with OD as response value” maybe better use in  table title: “Results of factorial regression model with …”.

Line 380

Maybe is possible to check in case of raw data if there are no any outliers, because, ie.: standard deviation for Direct Bilirubin in treatment 2.5% CU is higher than mean value, what emphasized very high variability of data. In this case  design of experiment and allocation of pigs to treatments from one litter will be important (50% to first and 50% to second treatment when even number of offspring).

Lines 653-783

References

           Dots are needed in every abbreviation of  Journal name: 3, 5, 9,

           Please check if all links for references works with first part of address: https://doi.org/ and display the article directly from website, ie. for reference no. 1: https://doi.org/10.3390/metabo12010063

           No. 10 space between Molecules and 2023 need to be added

           No. 18, 30 (italics and dots)

           No. 49 (Ann. Hepatol.)

Author Response

We sincerely thank the editor and reviewer for their valuable feedback that we have used to improve the quality of our manuscript. Please see the attachment.
